# A Review of Survey Instruments and Pharmacy Student Outcomes for Stress, Burnout, Depression and Anxiety

**DOI:** 10.3390/pharmacy12050157

**Published:** 2024-10-18

**Authors:** Kelly C. Lee, Austin Yan, Tram B. Cat, Shareen Y. El-Ibiary

**Affiliations:** 1UC San Diego Skaggs School of Pharmacy and Pharmaceutical Sciences, La Jolla, CA 92093, USA; 2Emory University, Atlanta, GA 30322, USA; austin.yan@emory.edu; 3UC San Francisco School of Pharmacy, San Francisco, CA 94143, USA; tram.cat@ucsf.edu; 4Department of Pharmacy Practice, College of Pharmacy, Midwestern University, Glendale Campus, Glendale, AZ 85308, USA; selibi@midwestern.edu

**Keywords:** scales, pharmacy student, burnout, stress, mental health

## Abstract

While the need to measure burnout, stress and mental health among pharmacy students has been emphasized in the literature, there is limited information on which validated scales should be used. The objective of this scoping review was to identify published studies that used validated scales for burnout, stress and mental health among pharmacy students to provide recommendations for implementation at schools/colleges of pharmacy. Thirty-two out of 153 articles published in the United States from 1 January 2000 to 30 September 2022 were included and categorized into studies measuring stress (20), burnout (4) and depression/anxiety (8). The most common validated scales used to assess stress and burnout among pharmacy students were the Perceived Stress Scale (PSS) and the Maslach Burnout Inventory and the Oldenburg Burnout Inventory, respectively. For mental health, anxiety was most commonly investigated using a variety of scales such as the Generalized Anxiety Disorder-7; the Patient Health Questionnaire, 9-item was used to measure depression in two studies. Validity, ease of use, cost and generalizability are important considerations for selecting a scale. The PSS has been studied extensively in pharmacy students and has been correlated with other well-being domains. Studies that measured burnout and mental health (specifically, depression and anxiety) have less published evidence among pharmacy students.

## 1. Introduction

Clinician well-being has gained national attention due to increased reports of clinician burnout and its impact on patient care and professional disengagement [1]. Specifically within the pharmacy profession, in 2019, several large and influential pharmacy organizations were convened to develop meaningful and actionable recommendations aimed at promoting resiliency in our pharmacy workforce [2]. In the academic pharmacy community, concerns surrounding student wellness and mental health issues have been increasing. In the 2017–2018 American Association of Colleges of Pharmacy (AACP) Student Affairs Standing Committee Report, members outlined charges to identify best practices to promote student well-being and resilience [3]. More recently, the AACP has made “Achieving Well-being for All” one of its top 2021–2024 strategic priorities [4]. As a result, there has been an impetus for pharmacy educators to evaluate factors that impact student burnout and implement strategies that can improve their well-being.

The National Academy of Medicine recently launched a National Plan for Health Workforce Well-Being in an effort to drive policy and systems change to strengthen the health workforce well-being [5]. The plan outlines several priority areas, but specifically to pharmacy students, it focuses on assessment tools, strategies and research, and the need to recruit and retain a diverse and inclusive health workforce. Unfortunately, there is no clear recommendation for how “well-being” should be measured in the workforce, nor in academic institutions.

There are many scales (validated and unvalidated) that are used by pharmacy schools to measure well-being. As discussed by Dyrbye and colleagues [6], several characteristics should be considered by organizations when selecting the appropriate scale [6]. Organizations should consider (1) the domain of well-being that is important to the stakeholder (in this case, the school/college of pharmacy), (2) the organization burden (i.e., brevity of scale, easy scoring, low or no cost), (3) actionable measures stemming from the scale (with easily to interpret scores and established benchmarks), (4) strong psychometric properties (validity and reliability) and (5) broad applicability (in the case of students, to all levels of students in different types of pharmacy programs) [6]. The domains of well-being are especially important since there are numerous definitions of well-being and schools/colleges of pharmacy should be selective in which construct of well-being they want to measure [6].

A recently published review of well-being assessment and interventions in pharmacy students included 15 articles (seven studies conducted in the US and eight conducted internationally) [7]. The review focused on studies assessing student pharmacist well-being (physical, economic, social, emotional, psychological, life satisfaction and engaging activities) using validated and non-validated scales specific for well-being. The authors also assessed for gaps in the literature on the topic and evaluated interventions for improvement. Due to the myriad definitions of well-being used by academic institutions and investigators as well as the issues of mental health challenges that are often associated with the topic of well-being, we aimed to provide a review of validated scales of stress, burnout and depression/anxiety that have been used to study pharmacy students in the United States. Our scoping review, therefore, is more extensive and encompasses a review of mental health challenges that have been frequently assessed in conjunction with student well-being. For each area, we organized the literature into common predictors, associations with other constructs and non-pharmacologic interventions. The main objective of the review was to provide pharmacy educators with the knowledge and validated tools that can be used to measure stress, burnout and depression/anxiety among their students.

## 2. Methods

One of the investigators conducted a systematic search of literature surrounding stress, burnout and depression/anxiety issues for pharmacy students using PubMed, Google Scholar and Google. We used the following search terms: pharmacy student, stress, burnout, mental health, mental wellbeing, perceived stress scale, PharmD, academic success, success predictors, Five Facet Mindfulness Questionnaire (FFMQ), Jefferson Scale of Empathy, Maslach Burnout Inventory (MBI), student survey and wellness scales. We also conducted secondary source searches using references found in the literature from the original search. We also received librarian assistance to identify additional publications using MeSH search terms: stress, psychological, burnout, pharmacy student and education.

Once the articles were identified and recorded using Microsoft Excel, two investigators independently reviewed the articles for inclusion or exclusion in the study. Any conflicts in the choice of articles were resolved upon discussion by the two investigators. Inclusion criteria consisted of English language articles that were published from 1 January 2000 to 31 August 2022 and articles that studied pharmacy students in the areas of stress, burnout and depression/anxiety disorders. Studies that included other types of students (e.g., medical) were included as long as pharmacy students were also studied. We excluded any studies that were conducted outside of United States due to curricular and student demographics that could be vastly different and impact outcomes related to stress, burnout and depression/anxiety. We also excluded any articles that (1) were review or editorial in nature, (2) did not use a validated scale to measure their outcomes (i.e., lack of validity studies in any sample), (3) were qualitative only (i.e., focus groups/interviews), (4) were peripherally related to well-being but topics studied were beyond the scope of this article (e.g., grit, alcohol use, illicit substance use) and (5) were duplicate.

The objective of this scoping review was to summarize the literature that focused on measuring stress, burnout and depression/anxiety among pharmacy students in the United States using validated scales. The review summarizes the scales that were used to measure each theme, describes the overall findings from the studies and provides overall recommendations for scales that may be the most predictive and feasible for implementation within schools/colleges of pharmacy. 

## 3. Results

In the systematic search, we initially identified 153 articles. Upon applying inclusion and exclusion criteria, 37 articles were selected for independent review by two investigators. The two investigators independently reviewed each article for inclusion or exclusion with a 95% (35/37) concordance rate. Upon resolution of conflicts between the two investigators, 32 articles were included in the systematic review (Figure 1).

We have summarized the scales that have been used in the studies along with their construct, scoring, accessibility and cost information (Table 1) [8,9,10,11,12,13,14,15,16,17,18,19,20,21,22,23,24,25,26,27,28,29,30,31,32,33]. The articles from the systematic search are summarized under the following themes: (1) stress, (2) burnout and (3) depression/anxiety. We identified 20 articles for stress (Table 2) [12,32,33,34,35,36,37,38,39,40,41,42,43,44,45,46,47,48,49,50], four articles for burnout (Table 3) [14,51,52,53], and eight articles for depression/anxiety (Table 4) [8,54,55,56,57,58,59,60]. For each theme, we provide a summary of published literature, including the scales used, major findings and limitations, and recommendations based upon validity, evidence and accessibility. Institutions should consider these factors in selecting the appropriate scale for their use. Modification of the validated scales is not recommended; however, institutions may combine the scales and/or add relevant programmatic questions regarding demographics and institution-specific offerings.

### 3.1. Stress

Stress is an emotional strain triggered by challenging factors like academic pressure, personal issues and environmental demands, especially in students. In our literature search, the most prevalent scale that was used to measure stress and reactions to stressful situations among pharmacy students was the Perceived Stress Scale (PSS 4, 10 or 14 item version) (Table 2). The PSS was used in 19 of 20 articles that were published after 2000 [12,32,33,35,36,37,38,39,40,41,42,43,44,45,46,47,48,49,50]. In one article, the Student-life Stress Inventory (SSI) was used instead of PSS to measure stress [34].

#### 3.1.1. Common Predictors of Stress

Among the studies evaluating stress and reported causes of stress, the commonly cited stressors for student pharmacists were academic course workload, high stakes activities such as examinations and reassessments, financial issues, and family and relationships [35,39,50]. In a study by Chisholm-Burns et al., investigators found that professional year 4 (P4) students who reported a greater “fear of debt” had also higher stress scores [41].

#### 3.1.2. Associations with Other Constructs

Student pharmacists’ perceived stress has often been evaluated in relation to their academic achievement and Health-Related Quality of Life (HRQOL). Study results by Maynor et al. [45,46] and Saul Ballard et al. [47] have shown that increased stress was significantly correlated with decreased academic self-concept (how well an individual feels they can learn). Marshall et al. [35] have also found students’ high stress levels to be negatively correlated with mental HRQOL (*p* < 0.001) [35].

When evaluating race/ethnicity factors related to stress, two studies reported no differences in stress level or stressors experienced by student pharmacists [12,40]. However, when evaluating stress levels in the different Doctor of Pharmacy (PharmD) school years, varying results have been reported [37,39,42,45,46]. While Awe et al. [12] found no difference in stress levels among class cohorts, Ford et al. [39] reported P2 students having the highest level of stress related to academic workload. In other studies, Garber et al. [42] reported P3 students being more likely to have higher levels of stress than P1 or P4 students, while on the contrary, Maynor and Baugh [45] reported that perceived stress was significantly lower in P3 students compared to P1 and P2 students. Some studies have reported increasing stress levels observed in students as they progress through the PharmD curriculum [36,44,49]. In the study by Spivey et al. [44], female and minority students particularly experienced greater levels of stress at orientation. Similar findings have also been reported by other studies where female students had higher perceived stress levels compared to male students [34,37,38]. In one study by Votta and Benau, students with lower GPAs had higher stress than those with higher grade point averages (GPAs) [37]. Further, Marshall et al. [35] found that the most significant stressor for both men and women was examinations, while Beal et al. [40] found that time spent with family/friends was the most frequently reported stress reliever [35,40].

To manage stress, student pharmacists often employed more positive coping strategies: spending time with family and friends, active coping, planning and acceptance, physical activity, meditation and yoga [12,35,40,42,45,46]. Interestingly, according to Garber et al., students who used exercise as a coping mechanism reported lower perceived stress (*p* < 0.01); meanwhile, students using maladaptive coping mechanisms (behavioral disengagement, venting and self-blame) were associated with higher perceived stress (*p* < 0.05) [42]. In studies by Hirsch et al., students observed to have maladaptive coping mechanisms were also found to have poorer mental health during their first three (pre-clinical) years of the four-year curriculum [36,49]. Unfortunately, maladaptive coping mechanisms such as use of alcohol and prescription or nonprescription drugs for anxiety or to aid sleep have been commonly cited as stress-relieving activities for students [35,38].

#### 3.1.3. Non-Pharmacological Interventions to Manage Stress

A few studies evaluated interventions to determine whether they were effective in helping students manage their stress. Lemay et al. conducted a 90 min yoga and meditation intervention and found that students’ Beck Anxiety Inventory (BAI) and PSS scores decreased significantly, and their Five Facet Mindfulness Questionnaire (FFMQ) scores increased significantly at the end of the six-week study period [58]. Similarly, Zollars et al. investigated the use of the Headspace™ app on mindfulness, mental well-being and perceived stress in pharmacy students [57]. Pharmacy students, who were in their P1, P2 and P3, were instructed to meditate at least 10 min daily for four weeks using the Headspace™ app. Study results showed that the intervention was associated with improved mindfulness, overall mental well-being and decreased perceived stress. Holman et al. [50] conducted a one-year pilot wellness program for P1 through P3 students, which consisted of the following interventions: orientation to the wellness program, session on nutrition and mindfulness, in-class brain breaks and promotion of on-campus resources. While the investigators found no statistical differences between pre- and post-PSS-10 scores for P1, P2, or P3 students, they did find that students had increased wellness practices in exercise and sleep (>4 h/night) (*p* = 0.02). Students also reported the greatest use of and satisfaction with 5–10 min in-class wellness breaks and provided the most feedback on curricular/schedule changes (e.g., reduced course load and rescheduling of campus wellness activities to fit into the course schedule) to improve student wellness.

### 3.2. Burnout

Our search revealed four studies matching our criteria to assess burnout in pharmacy students (Table 3). The Maslach Burnout Inventory (MBI) by Maslach et al. [13] was used in three of the studies while one utilized the Oldenburg Burnout Inventory (OLBI) [14,51,52,53]. Using the MBI, burnout is measured in three domains consisting of emotional exhaustion (EE), depersonalization (DP) and low personal accomplishment (PA) (Table 1). The OLBI uses two measures, disengagement and exhaustion. Using the MBI, Jacoby et al. [53] reported a prevalence of burnout in 82.3% of 62 third year pharmacy students (in a four-year program) and reported rates of high EE ranging between 46% and 85% of students in the first three years of pharmacy school [53]. For DP, high rates ranged from 25% to 33% depending on the year in school, and high rates of low PA were reported as 21% to 31% in the same study. Kaur et al. [52] and Ried et al. [51] reported average EE scores as 23 and 28, respectively, with a high score considered to be greater than 27 [51,52]. Per the MBI, a score of 7 to 12 in DP is considered moderate and 13 or greater is considered a high score and at risk for burnout [13]. The average range for DP reported in these studies was 6.7 to 15.9, showing low to high DP [51,52]. Also, per the MBI, low PA is defined as a score lower than 30. The average range reported for low PA in these studies was 13.5 to 8.9 [51,52]. The MBI scores show that pharmacy students in these studies were suffering from high EE and low PA, indicating possible burnout [51,52,53]. The OLBI showed average burnout scores of 45 (maximum score = 64) with disengagement having an average sum of 22 and exhaustion 23, resulting in the higher overall score of 45 [14]. Fuller et al. [14] reported mean scores of 2.7 for disengagement and 2.9 for exhaustion on a four-point scale. These scores for disengagement were cited by the authors as being higher than Greek (2.0 and 2.8), German (1.9 and 2.6) and medical residents (2.4 and 2.5) [14].

#### 3.2.1. Common Predictors of Burnout

There were a few common predictors of burnout that appeared in the different studies that included gender, year in school and campus location. According to Ried et al. [51] and Jacoby et al. [53], females reported more EE than males, with Jacoby et al. reporting that the EE score in females averaged 4.7 points higher than the depersonalization score [53]. Both studies, however, reported males scoring higher in depersonalization [51,53], with Jacoby et al. showing 70% of males reporting cynicism versus 41% of females [53].

The year in pharmacy school was also another common predictor. It seems the second year in pharmacy has consistently shown increases in measures related to burnout based on the available studies. Jacoby et al. conducted surveys at the start of the first year and end of the first three years of pharmacy school in a four-year program [53]. Of the students who completed all four surveys throughout the years, it was noted that burnout, which was defined as EE or cynicism in the study, increased after the start of pharmacy school and remained high through the years (*p* < 0.0001). Personal accomplishment scores were lower for the end of the first and second years compared to the start of the first year but there was no difference between the start of the first year and the end of the third year. Ried et al. reported that second-year students had significantly higher EE scores, by an average of 4.4 points, than first- or third-year students [51]. For PA, however, students in the second year scored higher than first-year students. Students further in the program perceived a greater lack of PA. Second-professional-year students were also more likely than first-year students to score higher on the DP domain [51].

Using the OBI, authors reported that second-year students had a 2.7 times higher risk of exhaustion than first-year students (95% CI, 1.2–6.2) [14]. Unlike exhaustion, disengagement risk was three times higher for second- and third-year students compared with first-year pharmacy students, (95% CI, 1.6–3.0) and (95% CI, 1.6–5.5), respectively. The same study also reported that unmarried students had a two times higher risk of exhaustion compared to married students (95% CI, 1.0–4.8), although this was not statistically significant.

Other findings by Jacoby et al. included students having higher EE levels when attending the main campus versus various satellite campuses [51]. When evaluating variables such as postgraduate plans, work involvement, extracurricular or co-curricular involvement, Fuller et al. did not find these variables to be significant predictors of student burnout [14].

#### 3.2.2. Associations with Other Constructs

Jacoby et al. [53] measured empathy using the Jefferson Scale of Empathy (JSE) along with the MBI starting at the beginning of the first year and then assessing at the end of each year in pharmacy school for three years. While burnout seemed to increase from the start of the first year, empathy remained the same throughout all years [53]. Kaur et al. [52] assessed burnout and engagement in P1 and P2 students using the MBI and the Utrecht Work Engagement Scale, and correlated scores to academic self-perception measured by a subscale of the School Attitude Assessment Survey-Revised. The study found that EE and low PA had a negative correlation with students’ academic self-perception (self-perceived academic abilities) [52].

#### 3.2.3. Non-Pharmacological Interventions to Manage Burnout

Studies assessing interventions for burnout in student pharmacists were not available. It is interesting to note that studies found similarities in terms of student characteristics leading to burnout. These include female gender and later year in school showing higher levels of EE (second and third years); higher levels of DP were also reported in males. In addition, low personal accomplishment scores were seen in the middle years of pharmacy school for most of the studies. Perhaps, interventions targeting these factors and populations may be of help as pharmacy schools work to improve overall well-being in students.

### 3.3. Depression and Anxiety

Our search generated eight articles that investigated pharmacy students’ depression and anxiety using validated rating scales (Table 4). There was no consistent scale or measurement used to measure students’ depression (consisting of low mood, anhedonia or other physical or cognitive symptoms) [59] or anxiety (i.e., feeling nervous, inability to control worrying, anticipatory anxiety) [59] in the published studies. The Patient Health Questionnaire, 9-item (PHQ-9) was used to measure depression in one study [59] while anxiety was measured using a variety of scales (i.e., the Generalized Anxiety Disorder, 7-item [GAD-7] [59], Beck Anxiety Inventory [58], Zung Self-Rating Anxiety Scale [60], State Trait Anxiety Inventory [STAI] [54]). Psychological or general mental health was measured using Counseling Center Assessment of Psychological Symptoms (CCAPS-62) in two studies [8,55] and Warwick-Edinburgh Mental Wellbeing Scale (WEMWBS) in one study [15].

Overall, pharmacy students in different cohorts have high levels of anxiety that meet clinical thresholds for anxiety diagnosis; Fischbein and Bonfine [56] found pharmacy students had higher rates of anxiety when compared to medical students [56]. In two studies by Shangraw et al. and Khorassani et al., second-year pharmacy students were shown to have the highest rate of anxiety compared to students in other years of study [59,60]. Students in second year (of four-year programs) tended to have high self-reported anxiety compared to those in other years, especially fourth-year students [59,60]. This is consistent with the findings from a study by Zakeri et al. [8] who found that female students were also shown to have higher anxiety than male students as well as decreased mental health [8]. In a study by Sabourin et al. [55], no association between mental health status and GPA or exercise was shown.

According to Geslani et al., pharmacy students not only had worse mental health compared to medical students, but they also had higher levels of stigma toward mental health treatment, were less likely to seek help from counseling services or know where to seek help [33]. Academic distress should be an area of focus for student support and intervention.

#### 3.3.1. Associations with Other Constructs

In two studies by Sabourin et al. and Fischbein et al., academic distress was shown to be correlated with depression and anxiety and moderately correlated with social anxiety [55,56]. High academic distress and family distress were associated with high clinical general anxiety [8].

#### 3.3.2. Non-Pharmacological Interventions to Manage Mental Health Disorders

There was little published information on interventions to reduce depression or anxiety among pharmacy students. Mindfulness meditation was associated with improved mental health, anxiety and stress [57,58].

## 4. Discussion

Student well-being is a growing concern within academic pharmacy because studies have found high levels of student pharmacist burnout associated with maladaptive coping mechanisms [36,49,53]. Multiple intrinsic and extrinsic factors, such as personal relationship and the culture and values of the learning environment, can influence student pharmacist stress and burnout [12,61,62]. As a result, there has been an increased interest in measuring well-being within pharmacy institutions.

The PSS has been validated in numerous populations to measure stress; therefore, this scale was frequently used as the measurement of choice for studies evaluating stress within pharmacy students (Table 2). Furthermore, this freely available scale with numerous versions (4, 10, 14 items) are easily accessible and scored and can be correlated with other areas that may be of interest for pharmacy schools (e.g., academic resilience) (Table 1). The SSI was also used in one study and the College Stress Inventory (CSI) scale was used alongside the PSS. Unlike the PSS, however, the high number of survey items as well as accessibility limit the frequent utility of these scales. Numerous studies showed that there are differences in perceived stress by gender and year in curriculum [37,41,44,47,48]; academic institutions may consider measuring perceived stress among their students at baseline and, possibly, every year to evaluate this pattern at their specific institution and design appropriate interventions. This regular assessment may also be helpful if there are curricular or programmatic revisions; the validated scale may potentially be used to measure the impact of these changes on students’ perceived stress. 

Surprisingly, despite the emphasis of burnout on wellness and well-being among healthcare professionals, there were very few studies using validated instruments to measure burnout among pharmacy students (Table 3). The potential reasons for this could be due to the accessibility and cost issues surrounding the gold standard scale for burnout, the MBI (Table 1). The MBI is only available through a copyright purchase agreement from MindGarden.com (accessed on 7 October 2024) and licenses are available for specific periods of time, which may not allow for easy implementation within academic institutions where repeated longitudinal assessments are necessary. While studies are limited, other burnout scales available may be an alternative solution, such as the OLBI, which is free for use (Table 1). The Copenhagen Burnout Inventory (CBI) is also another free instrument used to measure burnout. While both the OLBI and the CBI have no associated costs, there are no data available to compare the results to general population [6]. The CBI, to our knowledge, has also not been used in studies involving pharmacy students. The MBI, on the other hand, has national benchmark data that can be used to compare outcomes among pharmacy students to those among the general population and other samples. This type of normative data is not available for the OLBI or CBI. The burden to complete the burnout scales is similar, with the MBI having 22 items, the OLBI having 19 items, and the CBI having 16 items.

Mental health may be more challenging since there is a wide spectrum of domains that may encompass mental health. We chose to focus on depression and anxiety since these disorders have been highlighted as important factors among young individuals, especially in light of the COVID-19 pandemic [63]. We found that the PHQ-9 and GAD-7, both scales that are used clinically to screen for depression and anxiety, respectively, have been studied in pharmacy students. These scales may especially be desirable since they are easy to administer and score and are freely accessible (Table 1). The HRQOL scale may also be an alternate tool to measure overall health status since it is also relatively short and freely accessible from the Centers for Disease Control and Prevention [26].

### Limitations

Despite our attempt to be as thorough as possible in our literature search, there may be studies that could have been missed. Publications that are not in English or those that were not available in the selected databases could have been omitted.

## 5. Summary and Recommendations

There are numerous validated scales to measure well-being among pharmacy students and the selection of a specific tool may depend on the construct of well-being that institutions would like to measure. Critical factors such as validity, ease of use, cost and generalizability are also important considerations.

Based on the findings from the selected studies, pharmacy students are suffering from stress, burnout, depression and anxiety. Mental health and well-being are critical to the success of pharmacy students. Pharmacy schools should consider assessing student stress using the PSS-10, depression with PHQ-9 and anxiety using the GAD-7, particularly in the second year, where there were higher rates reported. Of note, the PSS has been studied extensively in pharmacy students and has been correlated with other domains of well-being. These tools are straightforward, free for use, easy to administer and will help identify struggling students who can benefit from support. Burnout may also be assessed if the institution has the funding to cover the cost of the MBI, which is the gold standard due to the normative data comparisons available. If cost is a concern, other burnout tools as mentioned may be used to obtain an idea of how students are feeling but using those instruments will have fewer data comparisons. Assessing burnout in the second and third years may be particularly helpful to institutions to determine whether burnout exists in their student population; if so, institutions may be able to implement programs or interventions to address these specific years where burnout seems most prevalent. Of available validated scales, the PSS-10, MBI, PHQ-9 and GAD-7 are the most published among pharmacy students and are viable options to assess stress, burnout, depression and anxiety, respectively.

## Figures and Tables

**Figure 1 pharmacy-12-00157-f001:**
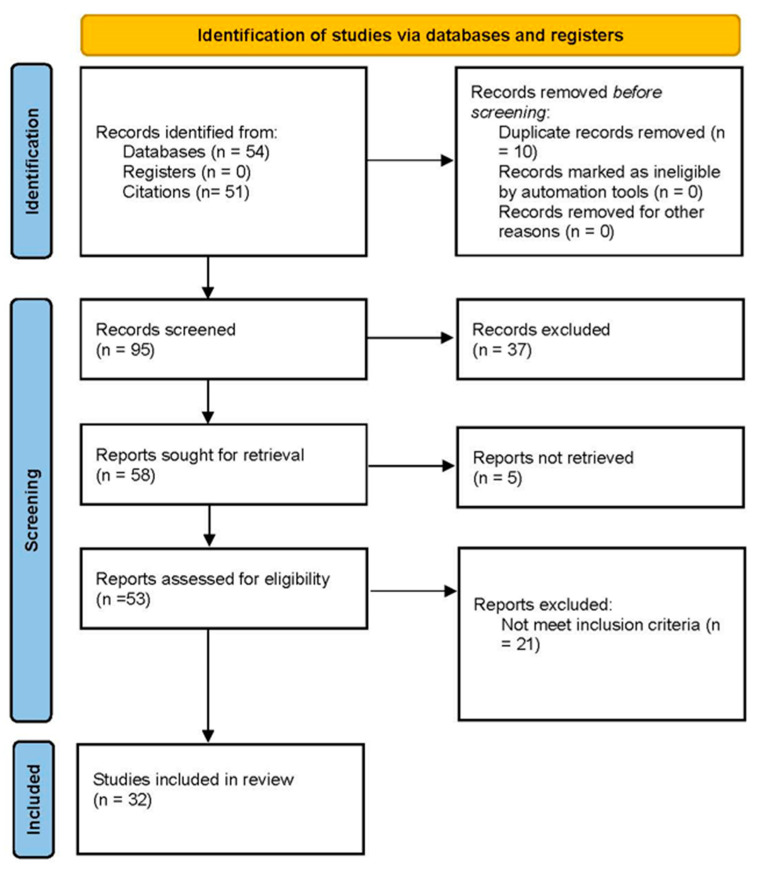
PRISMA flow diagram.

**Table 1 pharmacy-12-00157-t001:** Scales used in pharmacy programs to measure student well-being.

Scale	Construct	Number of Items	Score Interpretation	Copyright/Access/Cost
**Stress**				
Perceived Stress Scale (PSS) [8,9,10]	Stress	PSS-4: 4 PSS:10: 10 PSS-14: 14	Higher scores reflect higher stress	No cost or permission required when used for nonprofit academic research or purposes; for profit-making purposes, permission required from Journal of Health and Social Behavior (American Sociological Association) and the author (Dr. Cohen). https://www.cmu.edu/dietrich/psychology/stress-immunity-disease-lab/scales/ (accessed on 7 October 2024)
Student-life Stress Inventory (SSI) [11]	Stressors and reactions to stressors	51	Higher scores reflect higher stressors and reactions to stressors	No copyright or cost information. Contact Bernadette M. Gadzella (Texas A&M University Commerce).
College Stress Inventory (CSI) [12]	Stress (academic, social and financial sub-scales)	21	Higher scores reflect higher stress	No copyright or cost information. Contact V. Scott Solberg (ssolberg@bu.edu; Boston University Wheelock College of Education & Human Development).
**Burnout**				
Maslach Burnout Inventory (MBI) [13]	Burnout (emotional exhaustion, depersonalization, personal accomplishment)	22 16 (General Survey)	Scoring obtained from MindGarden.com	Online and paper versions available for purchase from www.mindgarden.com (accessed on 7 October 2024) Several versions available: Human Services General Educator Medical Personnel Students
Oldenburg Burnout Inventory (OLBI) [14]	Burnout (disengagement and exhaustion)	16	Higher scores reflect greater level of burnout (see reference for scoring instructions)	No cost to use.
**Depression/Anxiety**				
Warwick-Edinburgh Mental Wellbeing Scale (WEMWBS) [15]	Hedonic and eudaimonic aspects of mental health including positive affect, satisfying interpersonal relationships and positive functioning	14	Scored by summing all responses Range: 15–70 Higher scores indicate higher well-being	Validated for use in UK for those aged 16 years and above Freely available but prospective users should register with Dr. Kulsum Janmohamed (k.janmohamed@warwick.ac.uk or Professor Sarah Stewart-Brown sarah.stewart-brown@warwick.ac.uk).
Counseling Center Assessment of Psychological Symptoms (CCAPS-62) [16]	General anxiety, academic distress, family distress (Surveys 8 subscales of mental health)	62 (34 item version also available)	General Anxiety: 1.22–1.89, Academic Distress: 1.2–2.4, Family distress: 1.31–1.83	Developed by Counseling and Psychological Services at University of Michigan. Managed by Center for Collegiate Mental Health at PennState Student Affairs (https://ccmh.psu.edu/ccaps-34-62 (accessed on 7 October 2024)). Copyright and access information not available.
State Trait Anxiety Inventory (STAI) [17]	Adult anxiety	40 (Form Y-1 includes 20 items for state-anxiety, Form Y-2 includes 20 items for trait anxiety)	Range: 20–80 (39–40 suggested cut point for clinically significant state-anxiety) Higher scores reflect higher anxiety	Available for purchase from Mind Garden (http://www.mindgarden.com/index.htm (accessed on 7 October 2024)).
Patient Health Questionnaire, 9-item (PHQ-9) [18,19]	Depression	9	5–9, 10–14, 15–19, 20–27 Score of 10 or higher considered clinically significant	No permission required to reproduce, translate, display or distribute. https://www.apa.org/depression-guideline/patient-health-questionnaire.pdf (accessed on 7 October 2024)
Generalized Anxiety Disorder, 7-item (GAD-7) [20,21,22]	Anxiety	7	5–9, 10–14, 15–21 Score of 8 or higher considered clinically significant	No permission required to reproduce, translate, display or distribute.
Beck Anxiety Inventory (BAI) [23,24,25]	Self-report measure of anxiety	21	Range: 0–63 0–21: Low anxiety 22–35: Moderate anxiety 36 or higher: High anxiety	Copyrighted; available for purchase from Psychological Corporation, 555 Academic Court, San Antonio, TX 78204-2498, USA.
Depression, anxiety, and Stress Scale (DASS-21) [26,27,28]	Depression, anxiety and stress	21	Depression: 0–28+ Anxiety: 0–20+ Stress: 0–34+	No copyright or cost to use. https://maic.qld.gov.au/wp-content/uploads/2016/07/DASS-21.pdf (accessed on 7 October 2024)
Zung Self-Rating Anxiety Scale [29,30,31]	Anxiety	20	1 = insignificant intensity of anxiety 4 = severe anxiety Range: 20–80, score > 36 indicates clinically significant anxiety	Copyrighted. Cost and access information not available.
Medical Outcomes Study SF-12 and SF-36 Health Questionnaire [32]	Physical functioning, role functioning, social functioning, mental health, current health perceptions, pain	SF-12: 12 SF-20: 20 SF-36: 36	0 to 100 (higher scores reflect better functioning)	https://www.rand.org/health-care/surveys_tools/mos/20-item-short-form/more.html (accessed on 7 October 2024)
Health-Related Quality of Life Measures (HRQOL-14) [33]	Overall health status	14	Fourteen or more mentally unhealthy days = frequent mental distress (FMD) Higher scores correspond to poorer health	Free to use and publicly available from Centers for Disease Control and Prevention.

Note: Table is meant to be a general guide and should not be intended as sole authority with regards to scoring or copyright issues. Authors intending to use any of the scales should refer to the original validity studies as well as the author/developer of the scales.

**Table 2 pharmacy-12-00157-t002:** Studies in perceived stress.

Source	Population (Pharmacy, Medical, Class/Year)	Sample Size (Response Rate)	Instrument/Scale (e.g., MBI, PSS)	Results	Limitations (e.g., Small Sample Size, Single Institution, No Pre-Post, One Year of Data)
Gupchup et al., 2004 [34]	First-, second- and third-year pharmacy students	166 (88%)	Student-life Stress Inventory (SSI), Medical Outcome Study SF-12 Health Questionnaire (SF-12)	Strategies are needed to reduce student-life stress and improve the mental component of Health-Related Quality of Life (HRQOL)	Single institution, no pre-post data, one year of data, for same group
Canales-Gonzales et al., 2008 [32]	First-, second- and third-year pharmacy students	17 (100%)	Student Stress Questionnaire	70.5% reported using some form of active approach to stress reduction, demonstrating that students may benefit from programs that teach effective coping strategies	Small sample size, no pre-post data, one year of data
Marshall et al., 2008 [35]	Third-year pharmacy students	109 (83%)	Perceived Stress Scale (PSS)-14, Medical Outcomes Study SF-12 (SF-12), Health-Related Quality of Life Measures (HRQOL-14)	Students made mostly positive, but some negative, lifestyle choices to reduce stress	Single institution, no pre-post data; curriculum block scheduling not generalizable to other traditional semester schedules
Hirsch et al., 2009 [36]	First-, second- and third-year students	213 (67%)	SF-36, PSS-10, Brief COPE	Techniques to alleviate stress and/or reduce maladaptive coping skills are needed to improve students’ HRQOL throughout the pharmacy curriculum	Single institution, no pre-post data one year of data, volunteer participants resulting in self-selection bias
Geslani et al., 2013 [33]	First-year pharmacy students	139 (67%)	PSS-10, HRQOL-14	Women also reported poorer mental health than men and men reported fewer mentally unhealthy days; the most significant stressor for both men and women was examinations while spending time with family/friends was cited frequently as a stress reliever	Single institution, no pre-post data, one year of data, one class
Votta et al., 2013 [37]	First-, second-, third- and fourth-year pharmacy student members of American Pharmacists Association	2607 (16%)	PSS-10	Women and Asians reported higher stress than men and Caucasian students, respectively; GPA and year in program correlated negatively with stress.	Low response rate, one year of data, there may be geographic variations in the national sample that may not be generalizable
Frick et al., 2014 [38]	Second-year pharmacy students in a 3-year PharmD program	135 (70%)	PSS-14	Students in a 3-year PharmD program with a unique educational model experienced more stress than students in a traditional 4-year PharmD program	Single institution, no pre-post data, one class of students
Ford et al., 2014 [39]	First-, second-, third- and fourth-year pharmacy students	306 (60%)	PSS-14	High levels of perceived stress were found among PharmD students, which were mostly related to academic workload	Single institution, no pre-post data, one year of data
Beall et al., 2015 [40]	First-, second- and third-year pharmacy students	242 (68%)	PSS-10	Top stressors included class assignments and completing electronic portfolios; most frequent coping mechanism was spending time with family and friends	Single institution, no pre-post data, one year of data
Awé et al., 2016 [12]	First-, second- and third year pharmacy students at two universities with multiple campuses	820 (74%)	PSS-10, College Stress Inventory (CSI), Dental Environmental Stress questionnaire	Stress levels were similar at main or branch campuses	Single institution, no pre-post, one year of data
Chisholm-Burns et al., 2017 [41]	Graduating pharmacy students	147 (97%)	PSS-10, Attitudes-Toward-Debt Scale	Increased fear of debt was related to greater perceived stress, meaning that educational programming concerning loans, debt, and personal financial management may help reduce stress	Single institution, no pre-post data, one year of data
Garber, 2017 [42]	First-, second-, third- and fourth year pharmacy students	368 (81%)	PSS-10, Brief COPE scale	Perceived Stress score was 18.2 and differed by class cohort (*p* = 0.001) Third year students more likely to report higher levels of stress than P1 or P4 students Three maladaptive coping mechanisms (behavioral disengagement, venting and self-blame) were associated with higher perceived stress (*p* < 0.05) Those who reported using exercise as a coping mechanism reported lower perceived stress (*p* < 0.01)	Single institution, no pre-post data, one year of data
Garber et al., 2019 [43]	Pharmacy students across 3 different schools	352 (55%)	PSS-10	Most common predictors of high stress levels were academic performance (81%) and pressure to succeed (77%)	No pre-post data, one year of data, schools differed in curricular structure and length, low response rate
Spivey et al., 2020 [44]	First-year pharmacy students	201 (99.5%)	PSS-10	Higher perceived stress associated with lower academic performance. Accounted for school records, demographics, GPA	Single institution, one class of students, one year of data
Maynor et al., 2012 [45]	First-, second- and third-year pharmacy students	244 (unknown response rate)	PSS-14, ASCS, Brief COPE scale	PSS-14 scores were inversely related to ASCS scores Students reported high levels of perceived stress (mean 30.03, SD = 8.49) and perceived stress was significantly lower in the P3 class compared to P1 and P2 classes The academic self-concept score was high for the overall sample (mean score of 108, SD = 16.75) but significantly lower in the P2 class compared to P1 and P3 classes The most frequently used methods for coping were active coping, planning and acceptance	Unknown response rate, single institution, no pre-post data, one year of data
Maynor L et al., 2022 [46]	First-, second- and third-year pharmacy students	220 (92%)	PSS-14, Brief COPE scale, and ASCS	Perceived stress was reduced to a significant degree following a curricular revision Increased stress was statistically significantly correlated with decreased academic self-concept Students reported using self-distraction, but less frequently, active coping, substance abuse and planning for coping when compared to the previous cohort	Single institution, no pre-post data
Saul et al., 2021 [47]	First-, second- and third-year pharmacy students	113 (76%)	PSS-10, SF-12	No significant differences between native and non-native English speakers based upon three language-related criteria	Single institution, no pre-post data, one year of data
Verdone et al., 2021 [48]	First-year pharmacy, dental, medical and optometry students	404 (77%)	PSS-10, Basic Need Satisfaction Inventory	Female students showed higher perceived stress and lower basic need satisfaction Perceived stress levels did not differ by professional program (after accounting for gender) Higher basic need satisfaction was predictive of lower perceived stress in healthcare professional students Perceived stress levels remain higher for female students compared to male students	Single institution, no pre-post data, one year of data, one class of pharmacy students
Hirsch J et al., 2020 [49]	First-, second- and third-year pharmacy students	145 (46%)	PSS-10, Brief COPE scale, SF-36	There was declining mental health and life quality among pharmacy students as they progressed through pharmacy school	Single institution, low response rate
Holman et al., 2021 [50]	First-, second- and third-year pharmacy students	153 (92%)	PSS-10	One-year pilot wellness program No statistically significant difference between pre- and post PSS-10 scores for P1, P2, or P3 classes There was increased wellness practice in exercising and sleeping (>4 h/night) from pre- vs. post-implementation (*p* = 0.02) Students also reported greatest use of and satisfaction with 5–10 min in-class wellness breaks and suggested changes such as reduced course load, rescheduling of wellness activities to fit into course schedule	Single institution, one year of data

**Table 3 pharmacy-12-00157-t003:** Studies in burnout.

Source	Population (Pharmacy, Medical, Class/Year)	Sample Size (Response Rate)	Instrument/Scale (e.g., MBI, PSS)	Results	Limitations (e.g., Small Sample Size, Single Institution, No Pre-Post, One Year of Data)
Ried et al., 2006 [51]	Pharmacy students in main and three distance campuses	629 (91%)	Maslach Burnout Inventory (MBI)	Students at the founding campus in Gainesville reported more emotional burnout than students attending classes at the distance campuses	Single institution, no pre-post data, one year of data
Fuller et al., 2020 [14]	First-, second- and third-year pharmacy students	291 (75%)	Oldenburg Burnout Inventory (OLBI)	There was an increase in emotional exhaustion and disengagement among pharmacy students	Single institution, used OLBI making it harder to compare to other studies with MBI, no pre-post data, one year of data, timing of study was 3 weeks prior to final exams
Kaur et al., 2020 [52]	First- and second-year pharmacy students	361 (81%)	MBI, Utrecht Work Engagement Scale, Academic Self-Perception	Academic self-perception negatively correlated with emotional exhaustion and positively correlated with dedication	Single institution, no pre-post data, one year of data, students’ perceptions of academic ability (subjective measure)
Jacoby et al., 2021 [53]	First-year pharmacy students followed over three years	62 (69%)	MBI, Jefferson Scale of Empathy (JSE-MS)	Suggests high levels of burnout within PharmD students; empathy levels stayed relatively constant	Single institution, small sample size, JSE-MS designed for medical students, COVID-19 may have affected responses, study focused on third year students to assess effects of COVID-19

**Table 4 pharmacy-12-00157-t004:** Studies in depression and anxiety.

Source	Population (Pharmacy, Medical, Class/Year)	Sample Size (Response Rate)	Instrument/Scale (e.g., MBI, PSS)	Results	Limitations (e.g., Small Sample Size, Single Institution, No Pre-Post, One Year of Data)
Longyhore et al., 2017 [54]	Third-year pharmacy students	73 (87%)	40-item Spielberger State-Trait Anxiety Index (STAI)	The mean state-anxiety scores were higher than expected for a college student	Single institution, no pre-post data, one year of students, small sample size
Sabourin et al., 2018 [55]	First-, second-, third- and fourth-year pharmacy students and general university campus students	193 (58%)	Counseling Center Assessment of Psychological Symptoms (CCAPS)	PharmD students deal with higher levels of stress and, thus, are less mentally healthy; methods that reduce stress are necessary	Single institution, no pre-post data, one year data, low response rate
Fischbein et al., 2019 [56]	Pharmacy and medical students participating in the Healthy Minds Study	482 students (159 pharmacy students and 323 medical students) (response rate not reported)	Not specified	There are slight differences between depression/mental health/burnout rate between pharmacy and med students There are aspects that schools can improve to better the choice available for pharmacy and medical students	Exact response rate unknown due to weighting of samples, no pre-post data, one year of data
Zollars et al., 2019 [57]	First-, second- and third-year pharmacy students	92 (70%)	Five Facet Mindfulness Questionnaire (FFMQ), Warwick-Edinburgh Mental Well-Being Scale (WEMWBS), PSS	Mindfulness meditation correlated with improvement overall mental health	Single institution, quasi-experimental design, volunteer participation with incentive resulting in self-selection bias, personal reporting of meditation minutes
Lemay et al., 2019 [58]	First-, second- and third-year pharmacy students; undergraduate students	17 (85%) (9 in pharmacy program, 8 not in pharmacy program)	Beck Anxiety Inventory (BAI), PSS, FFMQ	Mean BAI and PSS scores decreased significantly after a 6-week study The FFMQ scores for acting with awareness and nonreactivity to inner experience increased significantly	Single institution, one year of data, small sample size
Shangraw et al., 2019 [59]	First-, second- and third-year pharmacy students	596 (82%)	Generalized Anxiety Disorder (GAD-7), PHQ-9 scales	More second-year pharmacy students self-reported anxiety and depressive symptoms as the semester progressed	Single institution, one year of data
Khorassani et al., 2021 [60]	First-, second-, third- and fourth-year pharmacy students	198 (18%)	Zung Self-Rating Anxiety Scale	Second-year students had higher reported anxiety whereas fourth year students had the lowest reported anxiety	Single institution, one year of data, low response rate
Zakeri et al., 2021 [8]	First-, second- and third-year pharmacy students	377 (63%)	CCAPS	Female students were more likely to have high clinical general anxiety than male students. Students who had high academic distress and high family distress had a higher probability of having high clinical general anxiety	Single institution, no pre-post data, one year of data

## Data Availability

No new data were created or analyzed in this study. Data sharing is not applicable to this article.

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
