# Peer review of "A Review of Survey Instruments and Pharmacy Student Outcomes for Stress, Burnout, Depression and Anxiety"

_pharmacy, 2024, doi:10.3390/pharmacy12050157_

Round 1
Reviewer 1 Report
Comments and Suggestions for Authors
Very nice job summarizing and well written. The problem I have is that it is slightly outdated and I am wanting more. The following are simple things that will greatly improve your paper:
1. Insert Prisma flow diagram for scoping reviews.
2. Given this is a review, I would do a search up to August 2024 to see if any additional data are out there (currently stops in 2022) and data will be > 2 years outdated by the time it is published. I would like to see a up-to-date article on this.
3. I know the focus is on the review of instruments. However, you discuss there are high rates of mental issues with students. Can you discuss within the articles listed what has been tried and if it worked or not. This will further strengthen your manuscript. I do not need a deep dive into this but some data on what has been tried and has it been effective or not. This section should not be more than a paragraph (keeping it simple)but it will provide some answers and your thoughts as to what else to do (or avoid) to serve pharmacy students.
Comments on the Quality of English Language
Just give it another readthrough but overall very well written.
Author Response
Comments 1: Very nice job summarizing and well written. The problem I have is that it is slightly outdated and I am wanting more. The following are simple things that will greatly improve your paper:
- Insert Prisma flow diagram for scoping reviews.
Response 1: Thank you. We have included the Prisma flow diagram (Figure 1)
- Given this is a review, I would do a search up to August 2024 to see if any additional data are out there (currently stops in 2022) and data will be > 2 years outdated by the time it is published. I would like to see a up-to-date article on this.
Response 2: Thank you for the comments. Unfortunately, expanding the review to 2024 will require significant revisions and would not be feasible at this time. We agree that a future article with expanded time frame should be published.
3. I know the focus is on the review of instruments. However, you discuss there are high rates of mental issues with students. Can you discuss within the articles listed what has been tried and if it worked or not. This will further strengthen your manuscript. I do not need a deep dive into this but some data on what has been tried and has it been effective or not. This section should not be more than a paragraph (keeping it simple)but it will provide some answers and your thoughts as to what else to do (or avoid) to serve pharmacy students.
Response 3: Thank you for your comment. While there are limited interventions that have been described in the literature cited, we did add details on the interventions that were reported in the Results section (unfortunately some of the text were missing from the submission and now has been included).
Reviewer 2 Report
Comments and Suggestions for Authors
The work is relevant and well-written. I think it can serve as a useful resource for pharmacy personnel and investigators who would like to identify appropriate scales for the assessment well-being in students. The work is very concise and straightforward in its delivery. I appreciate that as a technical reader.
Substantial amounts of data are presented in the tables. The tables do a pretty good job of conveying that information. However, the presentation of the data gets a bit unwieldy in the Anxiety/Depression section (pages 5 and 6). In particular, the statements presented in the Anxiety/Depression column run together. It is difficult to distinguish where one entry ends and the next entry begins. Formatting modifications are needed to fully distinguish the text entries. Table formatting can be a significant task, but the tables really drive this manuscript. It would be prudent to heavily emphasize them and make them easier to navigate.
Author Response
Comments 2: The work is relevant and well-written. I think it can serve as a useful resource for pharmacy personnel and investigators who would like to identify appropriate scales for the assessment well-being in students. The work is very concise and straightforward in its delivery. I appreciate that as a technical reader.
Substantial amounts of data are presented in the tables. The tables do a pretty good job of conveying that information. However, the presentation of the data gets a bit unwieldy in the Anxiety/Depression section (pages 5 and 6). In particular, the statements presented in the Anxiety/Depression column run together. It is difficult to distinguish where one entry ends and the next entry begins. Formatting modifications are needed to fully distinguish the text entries. Table formatting can be a significant task, but the tables really drive this manuscript. It would be prudent to heavily emphasize them and make them easier to navigate.
Response 4: Thank you. We have reformatted the table so that there is adequate spacing between the rows.
Reviewer 3 Report
Comments and Suggestions for Authors
This article titled “A Review of Survey Instruments and Pharmacy Student Outcomes for Stress, Burnout, Depression, and anxiety” is a scoping review which identified and categorized 32 articles. The study assessed which scales were mostly commonly used to assess stress, burnout, depression, and anxiety. Factors leading to selection of use included validity, cost, and generalizability. While the study offers valuable insights, its broader impact may be more focused in scope. The study is designed soundly and the manuscript is well written. Minor grammatical edits may improve readability.
Specific feedback is outlined below:
Abstract: I recommend explicitly stating the research question or objective.
Introduction: Inclusion of the research question and an explanation of why the research question lends itself to a scoping review approach would be useful.
Results: Inclusion of a flow diagram would be useful in outlining the search strategy including the specific phrase used in the search, number of articles meeting each exclusion criteria, and number of articles included. This supports reproducibility.
Page 3, line 111: I see reference to figure 1, but do not see this within the manuscript. Inclusion may address recommendations provided regarding flow diagram.
Table 1: Some inconsistencies were noted in the table. For example, most references are listed in brackets, while some are not. Some text appears gray. Some additional spacing between rows may improve readability (other tables were easier to read).
Page 16, lines 133-134: Add reference with which studies this refers to.
Page 17, lines 170-173: Recommend that the authors clarify why this is a limitation.
Page 17, lines 189-191: When recommending second and third year use, does this apply to 3-year programs? Clarification of why these years were noted would be helpful. Many studies in the table included first year students. Why was this year excluded from the recommendations?
Author Response
Comments 3: This article titled “A Review of Survey Instruments and Pharmacy Student Outcomes for Stress, Burnout, Depression, and anxiety” is a scoping review which identified and categorized 32 articles. The study assessed which scales were mostly commonly used to assess stress, burnout, depression, and anxiety. Factors leading to selection of use included validity, cost, and generalizability. While the study offers valuable insights, its broader impact may be more focused in scope. The study is designed soundly and the manuscript is well written. Minor grammatical edits may improve readability.
Specific feedback is outlined below:
Abstract: I recommend explicitly stating the research question or objective.
Response 5: Thank you. We have included the objective in the abstract
Introduction: Inclusion of the research question and an explanation of why the research question lends itself to a scoping review approach would be useful.
Response 6: Thank you. We have clarified the main objective of the scoping review in the Introduction. Previously, we have described the rationale for why the scoping review approach would be useful (Lines 61-77).
Results: Inclusion of a flow diagram would be useful in outlining the search strategy including the specific phrase used in the search, number of articles meeting each exclusion criteria, and number of articles included. This supports reproducibility.
Response 7: Thank you. We have included the Prisma flow diagram (Figure 1)
Page 3, line 111: I see reference to figure 1, but do not see this within the manuscript. Inclusion may address recommendations provided regarding flow diagram.
Response 8: Thank you. We have included the Prisma flow diagram (Figure 1)
Table 1: Some inconsistencies were noted in the table. For example, most references are listed in brackets, while some are not. Some text appears gray. Some additional spacing between rows may improve readability (other tables were easier to read).
Response 9: Thank you. We have attempted to correct the spacing issues between rows; however, we could not find where the references were not in brackets. We could not find any text that appeared gray – if you could let us know specific rows where this appears, we will correct. Thank you.
Page 16, lines 133-134: Add reference with which studies this refers to.
Response 10: Thank you. We have added the appropriate references.
Page 17, lines 170-173: Recommend that the authors clarify why this is a limitation.
Response 11: We omitted the sentences about not including the studies that did not use validated scales or pharmacy students since these are not limitations.
Page 17, lines 189-191: When recommending second and third year use, does this apply to 3-year programs? Clarification of why these years were noted would be helpful. Many studies in the table included first year students. Why was this year excluded from the recommendations?
Response 12: It seems that the some of the results section was missing from the original submission which described the role of the year in pharmacy school. We have included the details of why the recommendation of second and third year assessment was warranted.
Reviewer 4 Report
Comments and Suggestions for Authors
This manuscript aims “to provide a review of validated scales of stress, burnout, and depression/anxiety that have been used to study pharmacy students in the United States” (cited form lines 77-68 of the ms) and the authors choose to use the technique of writing a scoping review on this topic to achieve their aim. While the topic is of interest and relevant, and the authors have collected many references, this reviewer is of the opinion that the actual reviewing has not been done yet. The collection of the data is a good starting point, but a real analysis of the published data is required to be considered a scoping review. At this moment the manuscript only serves as a listing of published articles.
A scoping review, by a recent definition, is “a type of evidence synthesis that aims to systematically identify and map the breadth of evidence available on a particular topic, field, concept, or issue, often irrespective of source (i.e., primary research, reviews, non-empirical evidence) within or across particular contexts. Scoping reviews can clarify key concepts/definitions in the literature and identify key characteristics or factors related to a concept, including those related to methodological research” (Munn et al., JBI Evid Synth 2022 Apr 1;20(4):950-952, doi: 10.11124/JBIES-21-00483). It is, therefore, imperative that the present manuscript should result in a clear definition of the concepts ‘stress’, ‘burnout’, ‘depression’ and ‘anxiety’, as far as related to pharmacy students in the USA (their particular context).
Methodologically, the authors appear to have followed many required steps, but the description in the Methods section is somewhat unclear. What was the role of librarian assistance (only to identify additional serach terms)? It might be helpful for the authors to consult the PRISMA checklist for a scoping review (https://www.prisma-statement.org/scoping) to improve their manuscript.
It is necessary to make a clear disctinction between Results and Discussion; at the moment some results and some discussion topics are all mentioned in the Discussion section. In essence the manuscript lacks a description of Results, as needed in a scoping review (see PRISMA criteria). Under ‘Results’ the figure (referred to in line 111) is lacking and only references are made to the tables, but no critical appraisal of selected literature is presented. The reader would – at least – expect a clear defintition of the concepts ‘stress’, ‘burnout’, ‘depression’ and ‘anxiety’, and an analysis of the different scales, mentioned in Table 1. Making a disctinction between depression and anxiety is required. Are these scales measuring the same or different constructs? What is the overlap in scales? Can the authors make a selection of the best scale for making measurement in their context? At the end of the manuscript the authors make a choice for advising the PSS, MBI, PHQ-9 and GAD-7 for measurment of ‘stress’, ‘burnout’, ‘depression’ and ‘anxiety’, respectively (line 182-183), but this choice is not argued or based on a critical analysis of the data in the Results and/or Discussion section.
What are the criteria for inclusion of papers in the category stress (Table 2), burnout (Table 3) or depression/anxiety (Table 4)?
The manuscript might benefit from including a list of abbreviations.
This reviewer regrets to be soo critical, but it is hoped that the authors will be able to continue their analysis of the collected data and will be able to submit a thorough manuscript in the future. At the moment the manuscript must, unfortunately, be considered ‘immature’.
Author Response
Comment 4:
This manuscript aims “to provide a review of validated scales of stress, burnout, and depression/anxiety that have been used to study pharmacy students in the United States” (cited form lines 77-68 of the ms) and the authors choose to use the technique of writing a scoping review on this topic to achieve their aim. While the topic is of interest and relevant, and the authors have collected many references, this reviewer is of the opinion that the actual reviewing has not been done yet. The collection of the data is a good starting point, but a real analysis of the published data is required to be considered a scoping review. At this moment the manuscript only serves as a listing of published articles.
A scoping review, by a recent definition, is “a type of evidence synthesis that aims to systematically identify and map the breadth of evidence available on a particular topic, field, concept, or issue, often irrespective of source (i.e., primary research, reviews, non-empirical evidence) within or across particular contexts. Scoping reviews can clarify key concepts/definitions in the literature and identify key characteristics or factors related to a concept, including those related to methodological research” (Munn et al., JBI Evid Synth 2022 Apr 1;20(4):950-952, doi: 10.11124/JBIES-21-00483). It is, therefore, imperative that the present manuscript should result in a clear definition of the concepts ‘stress’, ‘burnout’, ‘depression’ and ‘anxiety’, as far as related to pharmacy students in the USA (their particular context).
Methodologically, the authors appear to have followed many required steps, but the description in the Methods section is somewhat unclear. What was the role of librarian assistance (only to identify additional serach terms)? It might be helpful for the authors to consult the PRISMA checklist for a scoping review (https://www.prisma-statement.org/scoping) to improve their manuscript.
It is necessary to make a clear disctinction between Results and Discussion; at the moment some results and some discussion topics are all mentioned in the Discussion section. In essence the manuscript lacks a description of Results, as needed in a scoping review (see PRISMA criteria). Under ‘Results’ the figure (referred to in line 111) is lacking and only references are made to the tables, but no critical appraisal of selected literature is presented. The reader would – at least – expect a clear defintition of the concepts ‘stress’, ‘burnout’, ‘depression’ and ‘anxiety’, and an analysis of the different scales, mentioned in Table 1. Making a disctinction between depression and anxiety is required. Are these scales measuring the same or different constructs? What is the overlap in scales? Can the authors make a selection of the best scale for making measurement in their context? At the end of the manuscript the authors make a choice for advising the PSS, MBI, PHQ-9 and GAD-7 for measurment of ‘stress’, ‘burnout’, ‘depression’ and ‘anxiety’, respectively (line 182-183), but this choice is not argued or based on a critical analysis of the data in the Results and/or Discussion section.
Response: We realized that the original submission was missing the Results section which addresses the comments above. We apologize for this gross error.
What are the criteria for inclusion of papers in the category stress (Table 2), burnout (Table 3) or depression/anxiety (Table 4)?
Response: The criteria for the included papers are listed in the Methods section (lines 89-102).
The manuscript might benefit from including a list of abbreviations.
Response: We have attempted to define all abbreviations within the text and the tables.
Editor – please let us know if a list of abbreviations is appropriate for addition and where it should be listed. Thank you.
This reviewer regrets to be soo critical, but it is hoped that the authors will be able to continue their analysis of the collected data and will be able to submit a thorough manuscript in the future. At the moment the manuscript must, unfortunately, be considered ‘immature’.
Response: We agree that the omission of the Results section severely limited the depth of the paper. We hope that the inclusion of the omitted text is appropriate.
Round 2
Reviewer 1 Report
Comments and Suggestions for Authors
Thank you for your revisions. I had only asked for a short paragraph but received a much larger well written comprehensive section. However, the time it took to revise this section would have been the time it would have taken to look up new studies in last 2 years and updated the table. This would have given you a modern up to date manuscript. I will defer to the editor.
Author Response
Comment 1: Thank you for your revisions. I had only asked for a short paragraph but received a much larger well written comprehensive section. However, the time it took to revise this section would have been the time it would have taken to look up new studies in last 2 years and updated the table. This would have given you a modern up to date manuscript. I will defer to the editor.
Response: The results section was missing from the original submission; therefore, we did not write this section de novo. We are still concerned with the time necessary to re-do a search and re-synthesize the literature. Thank you.
Reviewer 4 Report
Comments and Suggestions for Authors
The authors now supplied the Results section, which was missing on the original submission. This clarified many of the questions, which were the basis of my earlier review and reject advise. However, some of my original remarks have not yet been addressed by the authors:
Under ‘Results’ the figure (referred to in line 119) is lacking. This figure supposedly was intended to depict the manuscript selection process.
This manuscript aims “to provide a review of validated scales of stress, burnout, and depression/anxiety that have been used to study pharmacy students in the United States” (cited form lines 70-71 of the ms). The manuscript now contains a lot of data, related to the findings in the included references, ordered in a systematic way. This provides much information and a provides a helpful guide for readers when they consider studying aspect of stress, burnout, anxiety or depression among pharmacy students. What is still lacking, in my opinion, is a critical analysis or comparison of the scales used and an argued choice for advising the PSS (which version?), MBI, PHQ-9 and GAD-7 for measurment of ‘stress’, ‘burnout’, ‘depression’ and ‘anxiety’, respectively (line 190-192). Is their choice bases on availability only, on a critical analysis of the underlying psychological constructs? Can the published scales be used in the contet of pharmacy porgrammes in unmodifed form or are some modificatins necessary? Adding a few paragraphs in the Results section may be helpful.
The present manuscript illustrates even more that adding a list of abbreviations, or a summarizing description of the various psychological constructs (in table format) would increase readability of the manuscript.
The Results section uses two levels of subdivision (the psychological constructs, followed by description, predictors, associations and iterventions). Please take care making these levels clear in the layout and (sub)headings.
Author Response
Reviewer 4: The authors now supplied the Results section, which was missing on the original submission. This clarified many of the questions, which were the basis of my earlier review and reject advise. However, some of my original remarks have not yet been addressed by the authors:
Under ‘Results’ the figure (referred to in line 119) is lacking. This figure supposedly was intended to depict the manuscript selection process.
Response: The figure was uploaded separately but we have now included in the revised paper.
This manuscript aims “to provide a review of validated scales of stress, burnout, and depression/anxiety that have been used to study pharmacy students in the United States” (cited form lines 70-71 of the ms). The manuscript now contains a lot of data, related to the findings in the included references, ordered in a systematic way. This provides much information and a provides a helpful guide for readers when they consider studying aspect of stress, burnout, anxiety or depression among pharmacy students.
What is still lacking, in my opinion, is a critical analysis or comparison of the scales used and an argued choice for advising the PSS (which version?), MBI, PHQ-9 and GAD-7 for measurment of ‘stress’, ‘burnout’, ‘depression’ and ‘anxiety’, respectively (line 190-192). Is their choice bases on availability only, on a critical analysis of the underlying psychological constructs? Can the published scales be used in the contet of pharmacy porgrammes in unmodifed form or are some modificatins necessary? Adding a few paragraphs in the Results section may be helpful.
Response: We have added information in both Tables 2- 4 regarding the version of the PSS that were studied and in the Recommendation section for which version should be used.
We have defined stress (lines 159-160), burnout (lines 236-238), depression and anxiety (lines 315-317).
We have previously added our rationale for measurement recommendations including validity, ease of use, cost and generalizability (lines 174-175). We have added a few sentences regarding how these scales should be used in the Results section (lines 155-159).
The present manuscript illustrates even more that adding a list of abbreviations, or a summarizing description of the various psychological constructs (in table format) would increase readability of the manuscript.
Response: A table of abbreviations has now been included for readability. Table 1 lists all of the scales in a table format.
The Results section uses two levels of subdivision (the psychological constructs, followed by description, predictors, associations and iterventions). Please take care making these levels clear in the layout and (sub)headings.
Response: We have added the appropriate levels of subdivisions as indicated in the journal’s layout. The sublevels should now be clear with the respective subheadings.